biomechanics, evolution, neuroscience

insect, hearing, auditory fovea, sexual dimorphism, travelling wave

**Author for correspondence:**
Manuela Nowotny
e-mail: manuela.nowotny@uni-jena.de

# Comparative micromechanics of bushcricket ears with and without a specialized auditory fovea region in the crista acustica

Jan Scherberich[1,2], Roxana Taszus[1], Alexander Stoessel[1,3] and Manuela Nowotny[1,2]

[1]Institute of Zoology and Evolutionary Research, Friedrich-Schiller-University, Jena, Germany
[2]Institute of Cell Biology and Neuroscience, Goethe-University, Frankfurt am Main, Germany
[3]Department of Archaeogenetics, Max Planck Institute for the Science of Human History, Jena, Germany

JS, 0000-0003-1117-4287; RT, 0000-0003-3970-5270; AS, 0000-0003-2434-2542; MN, 0000-0002-2854-6908

In some insects and vertebrate species, the specific enlargement of sensory cell epithelium facilitates the perception of particular behaviourally relevant signals. The insect auditory fovea in the ear of the bushcricket *Ancylecha fenestrata* (Tettigoniidae: Phaneropterinae) is an example of such an expansion of sensory epithelium. Bushcricket ears developed in convergent evolution anatomical and functional similarities to mammal ears, such as travelling waves and auditory foveae, to process information by sound. As in vertebrate ears, sound induces a motion of this insect hearing organ (crista acustica), which can be characterized by its amplitude and phase response. However, detailed micromechanics in this bushcricket ear with an auditory fovea are yet unknown. Here, we fill this gap in knowledge for bushcricket, by analysing and comparing the ear micromechanics in *Ancylecha fenestrata* and a bushcricket species without auditory fovea (*Mecopoda elongata*, Tettigoniidae: Mecopodinae) using laser-Doppler vibrometry. We found that the increased size of the crista acustica, expanded by a foveal region in *A. fenestrata*, leads to higher mechanical amplitudes and longer phase delays in *A. fenestrata* male ears. Furthermore, area under curve analyses of the organ oscillations reveal that more sensory units are activated by the same stimuli in the males of the auditory fovea-possessing species *A. fenestrata*. The measured increase of phase delay in the region of the auditory fovea supports the conclusion that tilting of the transduction site is important for the effective opening of the involved transduction channels. Our detailed analysis of sound-induced micromechanics in this bushcricket ear demonstrates that an increase of sensory epithelium with foveal characteristics can enhance signal detection and may also improve the neuronal encoding.

## 1. Introduction

In acoustically communicating animals, sound production organs and sound receiving organs coevolve under the pressure of natural (e.g. eavesdropping predators and parasites) and sexual selection (e.g. female choice) [1–3]. In cluttered habitats like tropical rainforests, often a multitude of acoustic animals (different species of insects, frogs, birds and mammals) compete for communication niches by adapting their signalling characteristics like frequency spectrum, calling distance and activity times [4–6]. Sound signals (sender) and auditory signal perception (receiver), however, need to stay attuned [7–9].

In the Malaysian bushcricket *Mecopoda elongata* (Tettigoniidae: Mecopodinae) and most other orthopterans, the males produce a species-specific calling song so that conspecific females can phonotactically find them [10,11]. Bushcrickets of the

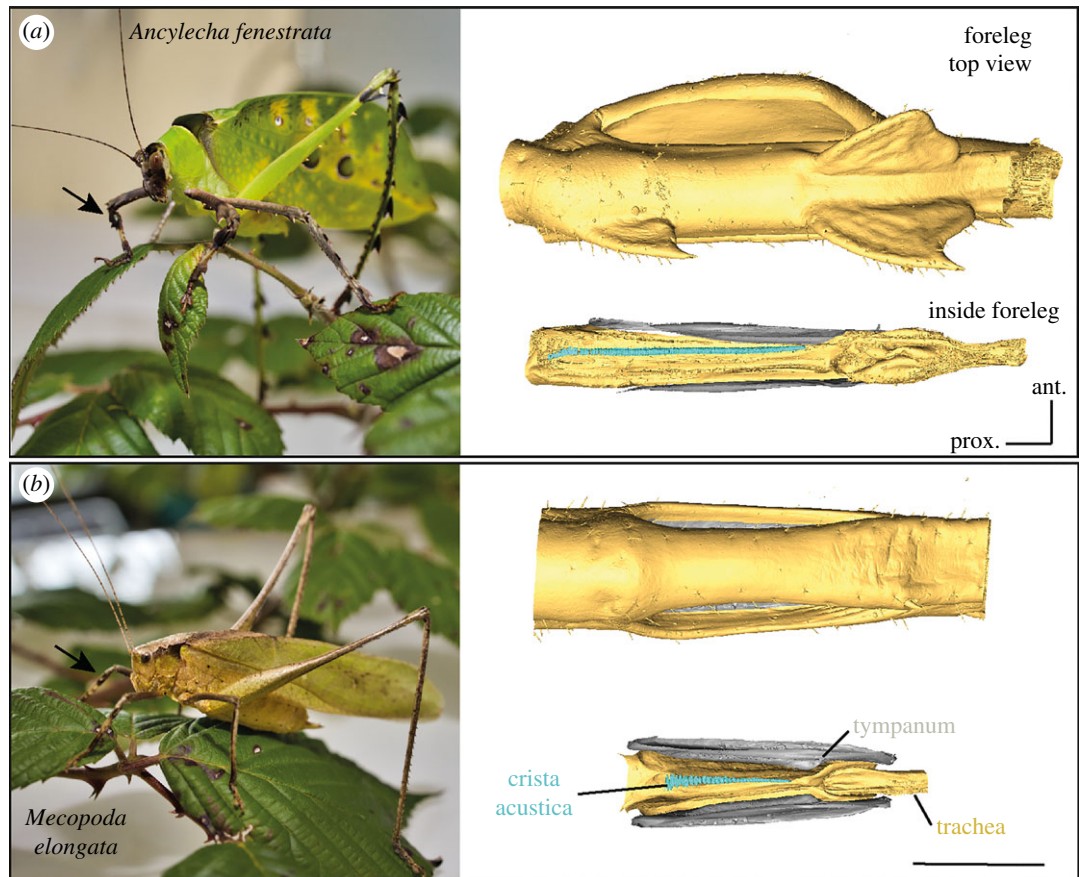

**Figure 1.** Ears of the bushcricket species *Ancylecha fenestrata* and *Mecopoda elongata*. (*a,b*) Photograph of male specimen (left side; black arrow indicates location of the ear in the foreleg tibia) and µCT-scan of the foreleg in the region of the hearing organ with the crista acustica (right side) in *A. fenestrata* (*a*) and *M. elongata* (*b*). Digital removing of the surrounding cuticle (yellow) from the µCT top view (top) reveals an inside view (below) of the leg trachea in yellow and crista acustica with sensory cells in blue. Both tympana are visible in grey. In the upper graph from *A. fenestrata*, the laterally tympana are hidden by the leg cuticula. Scaling bar, 1 mm. (Online version in colour.)

subfamily Phaneropterinae (Orthoptera: Tettigoniidae) are an exception, in which males and females form an acoustic duet as a mate-finding strategy [12,13]. To answer the male calls, females independently developed their own stridulatory organs (file and scraper structure) on the forewings, which slightly differ from those reported in males [14,15]. The female stridulatory organs always consist of small teeth on the dorsal surface of the right elytron (front wing protecting the hind wings) but show distinct variation among different species [16]. Using a duetting strategy, females respond to the male call with a very short sound pulse within a specific time window to reveal their location to the phonotactically searching male [13,15,17]. The sound spectrum of the male calls is usually broadband, while the female responses are often rather narrowband [16,18]. Low-frequency components of a broadband signal can enhance the calling distance [6] in the tropical rainforest, whereas narrowband calls could improve acoustic competition with sympatric species [8].

In the tropical Phaneropterinae species *Ancylecha fenestrata*, the animals show a different dominant sound frequency in male (about 30 kHz) and female (about 10 kHz) calls and also sex-specific differences in the morphology and physiology of the tonotopically organized crista acustica in their sound receiving ears [16,19]. These ears can be found in the forelegs of the animals (figure 1). Males possess a specialized region along the crista acustica with a pronounced overrepresentation of the behaviourally important frequency range at about 10 kHz, which fits the dominant frequency of the female call.

Such sexual dimorphism in the sensory structures is unusual in tympanal ears of insects (e.g. [20,21]). However, a similar frequency overrepresentation, called auditory foveae, can be found in the cochlea of some vertebrates such as bats [22], barn owls [23], mole rats [24] and the kiwi [25] as adaptation to the challenges of acoustic orientation behaviours. Based on the original findings in bats, Kössl *et al*. [26] described three criteria of an auditory/acoustic fovea: (i) an expanded cochlear representation of a sensitive narrow frequency range, (ii) an enhanced sharpness of neuronal tuning for the overrepresented characteristic frequency in this region and (iii) an increased density of nerve fibres that connects to the inner hair cells to process the signal. In vertebrates with an auditory fovea, only the ears of bats fulfil all three criteria [26]. In the bushcricket *A. fenestrata*, the first two criteria are met [16,19]. Bushcrickets possess bipolar primary sensory cells in their ears. Therefore, a higher density of nerve fibres in the ear is an unfeasible criterion in these ears. Unlike some other animals with sexually dimorphic ears [27] *A. fenestrata* males and females are sensitive to the same overall frequency range [16].

Like mammal ears [28], bushcricket ears are tonotopically organized [29]. In both cases, the frequency-discrimination properties of the ears are based on the mass and stiffness gradients along the hearing organ. A halt in the anatomical gradient (e.g. organ height) in the foveal region of bushcricket and also bat hearing organs lead presumably to constant mass and stiffness in this region. In the females of *A. fenestrata*, this

gradient suspension is less pronounced and therefore the auditory fovea is not as prominent as in males. This sex-specific difference is most evident from a shorter organ length and fewer sensory units in the female ear [19]. So far, the biomechanics of this fovea region are still unknown. A foveal standing wave in the bushcricket ear, as found in mammals [30], would contradict our previous findings in *M. elongata*, which indicate the need of a phase change as mechanical basis of signal transduction. In general, oscillatory organ motion in bushcrickets can be described by an amplitude and phase component. When in the direction of the mechanical wave propagation, the phase response lag along the hearing organ is called a phase delay. It has been shown that tilting of the cap cell and connected ciliated tip of the sensory dendrite, resulting from the phase delay of the travelling waves, is necessary to open the transduction channels [31]. Therefore, the question arises how these phase delays in the bushcricket ear are affected when the gradient of stiffness change is interrupted in the auditory fovea. The knowledge gained can help to understand how ear micromechanics are influenced in other taxa with an auditory fovea. Using laser-Doppler vibrometry and μCT data, we investigated this question by a comparative analysis of the micromechanics in bushcricket ears without (*M. elongata*) and with an auditory fovea (*A. fenestrata*).

## 2. Material and methods

### (a) Animals and preparation

Experiments were performed using the two tropical bushcricket species *Ancylecha fenestrata* (Tettigoniidae: Phaneropterinae; male = 9, female = 13) and *Mecopoda elongata* (Tettigoniidae: Mecopodinae; male = 9, female = 9) from our breeding colony at the Department of Cell Biology and Neuroscience in Frankfurt am Main (Germany). Additional *A. fenestrata* specimens were obtained from European breeders at larval stage. For measurements of the mechanical crista acustica responses, we used the dissection technique previously described in detail (e.g. [32,33]). Briefly, the animal was fixed onto a holder and the tibia of a foreleg was mounted in stretched position within a small fluid-filled chamber. The cuticle above the crista acustica was carefully removed and during paused measurements the crista acustica was regularly moistened with fresh Ringer's saline for insects [34].

### (b) Micro-computed tomography

To investigate the anatomical structure of bushcricket ears, μCT data were obtained using the Skyscan 2211 X-ray nanotomograph (Bruker, Belgium) at the Max Planck Institute for the Science of Human History in Jena. The forelegs of two male individuals from each species were dehydrated in an ethanol series and iodine stained to maximize contrast. Subsequently, they were dried at critical point (EmiTech K850 Critical Point Drier) shortly before scanning with a beam strength of 50 kV for *Mecopoda elongata* and 60 kV for *Ancylecha fenestrata* and exposure time of 2200 and 3100 ms, respectively. The achieved image spatial resolution of 9.05 mm voxel size allowed for detailed study of the hearing organ structures with imaging software (Avizo 2019.1, Visualization Science Group) to create 3D isosurface models either adjusted to a single value threshold or by manual segmentation for finer structural elements such as the sensory cells.

### (c) Laser-Doppler vibrometry and sound stimulation

To measure in detail the displacement amplitude and phase response of the crista acustica in response to acoustical stimuli up to 20 kHz, a microscanning laser-Doppler vibrometer system (MSV-300 with a sensor head OFV-534; Polytec) was attached to an upright microscope (Axio Examiner A1; Zeiss). The laser beam was adjusted by video control (VCT-101; Polytec). A line of measuring points every 5–16 μm was placed on top of the sensory cells. The total number of measuring points depend on the length of the hearing organ: *A. fenestrata*: 75–123, *M. elongata*: 43–91. We started the experiments in *M. elongata* with a lower number of measuring points (43, *n* = 3) and increase this number later for a higher resolution (91, *n* = 15). Each point measurement was repeated 20 times and averaged (128 kHz sampling rate). Displacement responses were analysed with a fast Fourier transformation (50 kHz bandwidth, 1600 lines, 31.25 Hz resolution, Hanning window). Phase values were obtained in relation to the sound stimulus. For acoustic stimulation, pure tones from 2 to 20 kHz (0.5 kHz steps) were produced by a function generator (NI 611x; Polytec), adjusted in SPL (audio attenuator 350D; HP Company) and amplified (RB-850; Rotel) to drive a broadband speaker (R2904/700000; ScanSpeak) that was placed about 30 cm from and perpendicular to the ipsilateral spiracle, such that the SPL at the spiracle was 80 dB SPL. Sound pressure levels across the frequency range were calibrated using a condenser microphone (MK301, Microtech) that was calibrated beforehand (sound calibrator type 4231 and measuring amplifier type 2610, Brüel & Kjær).

### (d) Data analysis and statistics

Amplitude and phase response data of all animals were pre-processed using custom-written MATLAB scripts (MATLAB 2019b, Mathworks) for phase unwrapping. These data of each of the four groups (*A. fenestrata* male/female and *M. elongata* male/female) were averaged and smoothed (moving average: five data points). For all measurements, a line layout of scanning points adjusted to sensory unit position was used. With a variation below 10 μm, data points along the crista acustica were fitted between different animals for averaging. To quantify the area that is affected by the wave amplitude, we calculated the area under curve (electronic supplementary material, figure S1A) with the mechanical displacement amplitudes and ear length using MATLAB routines (MATLAB R2019b, Mathworks). Broadness of mechanical response curves (electronic supplementary material, figure S1B) was calculated 3 dB below peak position for each stimulus frequency as percentage of the crista acustica length using MATLAB (MATLAB R2019b, Mathworks). Statistical analysis was performed using PAST 3.25 [35]. Amplitude and phase data were compared between both species and sexes by two-way ANOVA.

## 3. Results

To determine fovea-related biomechanical characteristics of the ears, we played pure-tone sound stimuli in 0.5 kHz steps from 2 kHz up to 20 kHz and measured the displacement amplitude and phase responses of the crista acustica in bushcrickets with (*Ancylecha fenestrata*) and without (*Mecopoda elongata*) an auditory fovea that belong to different subfamilies (Phaneropterinae and Mecopodinae, respectively). Because of the sexual dimorphism in the crista acustica of *A. fenestrata*, we analysed for both species the data of each sex separately.

Crista acustica length in *A. fenestra* males was approximately 1.9 mm and in females 1.6 mm. In *M. elongata*, in both sexes, the crista acustica is only about 1.0 mm long (figures 1 and 2). In both species and sexes, there was a clear tonotopic distribution of the displacement maxima in response to different sound frequencies along the proximo-distal axis of the crista acustica (figure 2). For low-frequency sound, the amplitude maxima of the travelling waves occurred in the

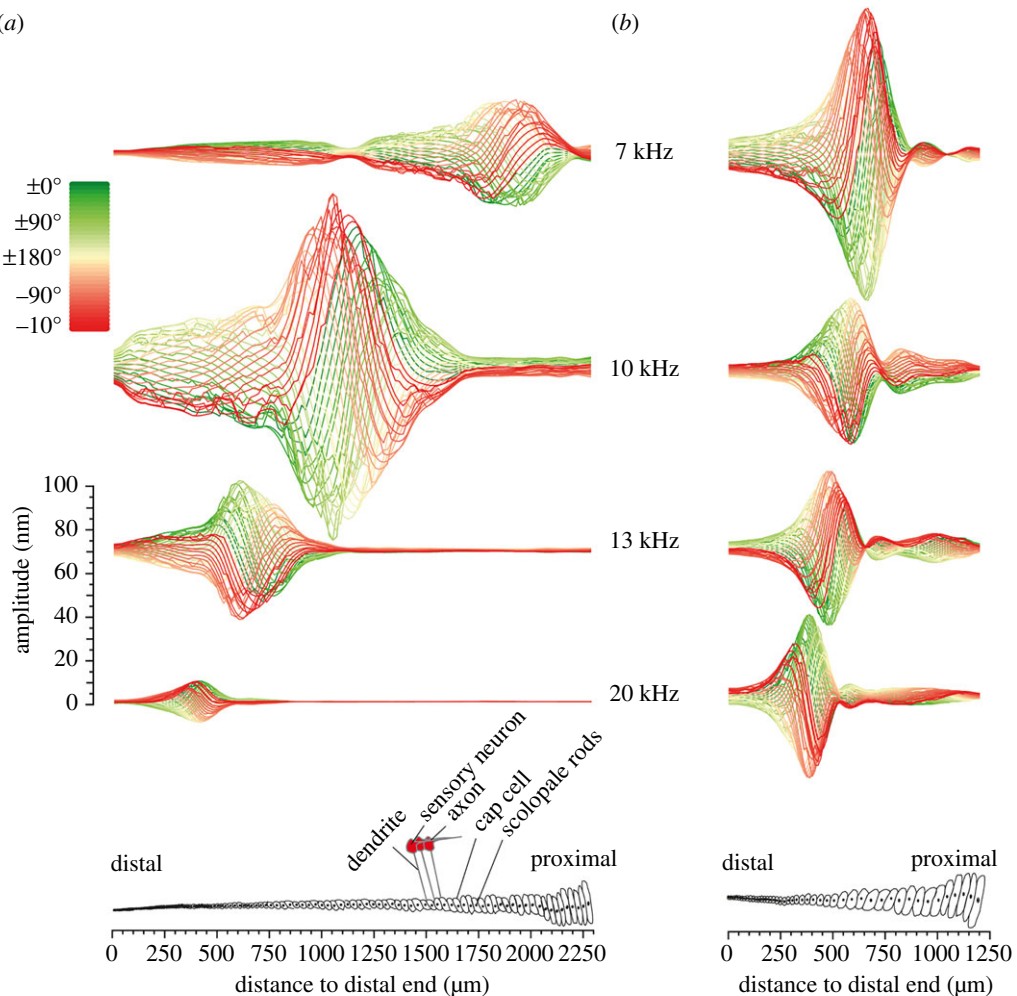

**Figure 2.** Micromechanics of the crista acustica in *Ancylecha fenestrata* and *Mecopoda elongata*. (*a,b*) Laser-Doppler vibrometry measurements of mechanical responses along the crista acustica for stimulation with sound of different frequencies at 80 dB SPL are shown for one recording example of a male *A. fenestrata* (*a*) and a male *M. elongata* (*b*) specimen. Below, a schematic top view of the crista acustica for each species is showing the corresponding organ shape. The colour code displays a full oscillatory cycle with phase angles from 0° to 180° (green) and phase angles from −180° back to 0° (red). In both species, the frequency-specific response maxima showed a clear tonotopic distribution along the hearing organs with low-frequency representation in the proximal part to high-frequency representation in the distal part of the crista acustica. In these examples, the highest response amplitudes were recorded in the medial part of both ears with 10 kHz stimulation for *A. fenestrata* (*a*) and 7 kHz stimulation for *M. elongata* (*b*). (Online version in colour.)

proximal part of the crista acustica and for high-frequency sound in the more distal part of the hearing organ. In the males of *A. fenestrata*, highest displacement amplitudes were typically generated for sound frequencies around 10 kHz in the middle of the crista acustica (figure 2*a*). In females, highest deflections occurred around 10.5–15.5 kHz with the maximum found at 15.5 kHz after the first third to the distal end of the crista acustica (for details please table 1). Mechanical displacement maxima in *M. elongata* (males and females) peaked for about 12–13 kHz with about 0.5 mm distance to the distal end.

Comparing the average deflection values between sexes in *A. fenestrata* (species with a sex-dimorphic auditory fovea) we found along the crista acustica significantly higher displacement amplitudes in male ears for sound frequencies from 9.0 to 10.5 kHz (figure 3*a,b*). The highest displacement amplitude in *A. fenestrata* males was recorded for 9.5 kHz sound stimulation ($39 \pm 32$ dB rel. nm, $n = 9$; for nm values please see table 1). In *A. fenestrata* females, we recorded the highest deflection amplitudes ($35 \pm 30$, dB rel. nm, $n = 14$) in response to 15.5 kHz sound stimuli. The phase delay at the deflection maximum of the sound-induced motion was $-219° \pm 37°$ in males ($n = 9$) and $-190° \pm 43°$ in females ($n = 14$). Normalizing the start of motion at the distal end of the crista acustica, these

phase values correspond to the delay of motion up to the point of maximum deflection. Higher phase delays are reached when the total sound-induced motions along the crista acustica is analysed (figure 3, phase delay at highest amplitude = coloured stars, highest phase delay of overall motion = triangles, for details, see table 1). In the male ears of *A. fenestrata*, these total phase delays were higher than $-270°$ for frequencies from about 8–13 kHz (figure 3, red shaded area). The maximum phase delays (figure 3) were recorded in males for 9.5 kHz sound stimulation with $-408° \pm 93°$ (amplitude: $12 \pm 8$ dB rel. nm) and in females for 9.0 kHz sound with $-327° \pm 100°$ (amplitude $12 \pm 7$ dB rel. nm).

In the ears of *M. elongata* (species without auditory fovea), there is a tonotopic amplitude gradient present as well (figure 2). The mechanical response of the crista acustica to pure-tone stimuli showed no sex differences (figure 4; $p = 0.775$, two-way ANOVA). With a total organ length of about 1 mm in both sexes, the highest deflection amplitudes were reached with $30 \pm 34$ dB rel. nm in males ($n = 9$) and $33 \pm 29$ dB rel. nm in females ($n = 9$) equally for 12.5 kHz sound stimulation. These amplitude values were significantly lower ($p < 0.001$, two-way ANOVA) compared to the highest deflection amplitudes in *A. fenestrata* males in response to 9.0–10.5 kHz

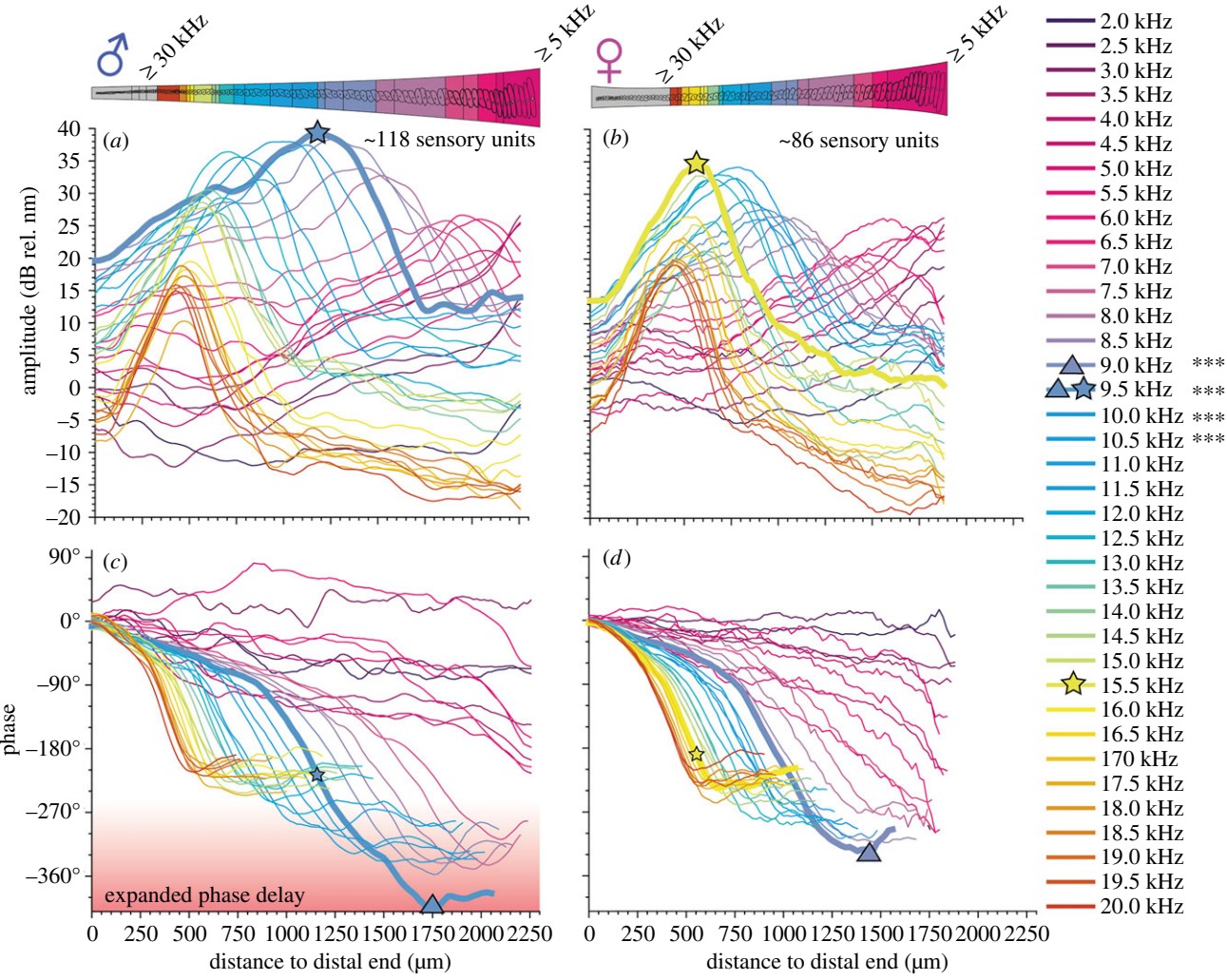

**Figure 3.** Mechanical responses of the crista acustica to pure-tone stimulation in male and female *Ancylecha fenestrata*. Mean mechanical response to pure tones of 80 dB SPL from 2 to 20 kHz (0.5 kHz steps, see colour coded legend) was measured in male (*a,c*, n = 9) and female (*b,d*, n = 14) ears of *A. fenestrata*. Male and female ears have different length (mean values: 2.25 mm and 1.8 mm, respectively) and numbers of sensory cells (mean values: 118 and 86, respectively) as shown in the sketch above. Data revealed the tonotopic gradient in the response amplitude (*a,b*) and phase delay (*c,d*). Highest response amplitude (thick lines), were found at 9.5 kHz in males and about 15.5 kHz in female ears. In the male ears, pronounced phase delays of more than −270° were measured for frequencies from about 8 kHz to 13 kHz (red shaded area of higher phase delays compared to females of *A. fenestrata* or males and females of *M. elongata*). The position of the highest amplitude response is marked by large stars and highest phase shift frequency and position is marked by a large triangle (frequency marked in the legend). Small stars indicate the phase delay at the position of highest organ deflection. The response amplitudes were significantly higher in males at 9–10.5 kHz in comparison to the responses in female ears (***p < 0.001, two-way ANOVA). (Online version in colour.)

**Table 1.** Characteristics of the mechanical response at maxima of deflection and phase delay. Values correspond to the coloured stars/triangles in figures 3 and 4.

| | parameter | male *A. fenestrata* | female *A. fenestrata* | male *M. elongata* | female *M. elongata* |
|---|---|---|---|---|---|
| for deflection maximum (large stars) | sound frequency | 9.5 kHz | 15.5 kHz | 12.5 kHz | 12.5 kHz |
| | defection amplitude | 91 ± 41 nm | 55 ± 31 nm | 32 ± 48 nm | 43 ± 29 nm |
| | phase delay (small stars) | −219° ± 37° | −190° ± 43° | −127° ± 60° | −166° ± 31° |
| | position from distal (% of organ length) | 1.2 mm (55%) | 0.6 mm (33%) | 0.5 mm (42%) | 0.5 mm (42%) |
| for phase delay maximum (large triangles) | sound frequency | 9.5 kHz | 9.0 kHz | 19.5 kHz | 12 kHz |
| | defection amplitude | 3.8 ± 2.5 nm | 3.9 ± 2.3 nm | 1.6 ± 2.0 nm | 2.5 ± 2.6 nm |
| | phase delay | −408° ± 93° | −327° ± 100° | −311° ± 178° | −323° ± 122° |
| | position from distal (% of organ length) | 1.7 mm (78%) | 1.4 mm (64%) | 0.6 mm (50%) | 0.8 mm (67%) |

sound. The phase delays at the point of highest deflection (12.5 kHz sound) was −127° ± 60° (n = 9) in *M. elongata* males and −166° ± 31° (n = 9) in females (table 1).

Relative broadness of the deflection amplitude curves was highest in *A. fenestrata* (with no difference between males and females) for sound stimulation up to about

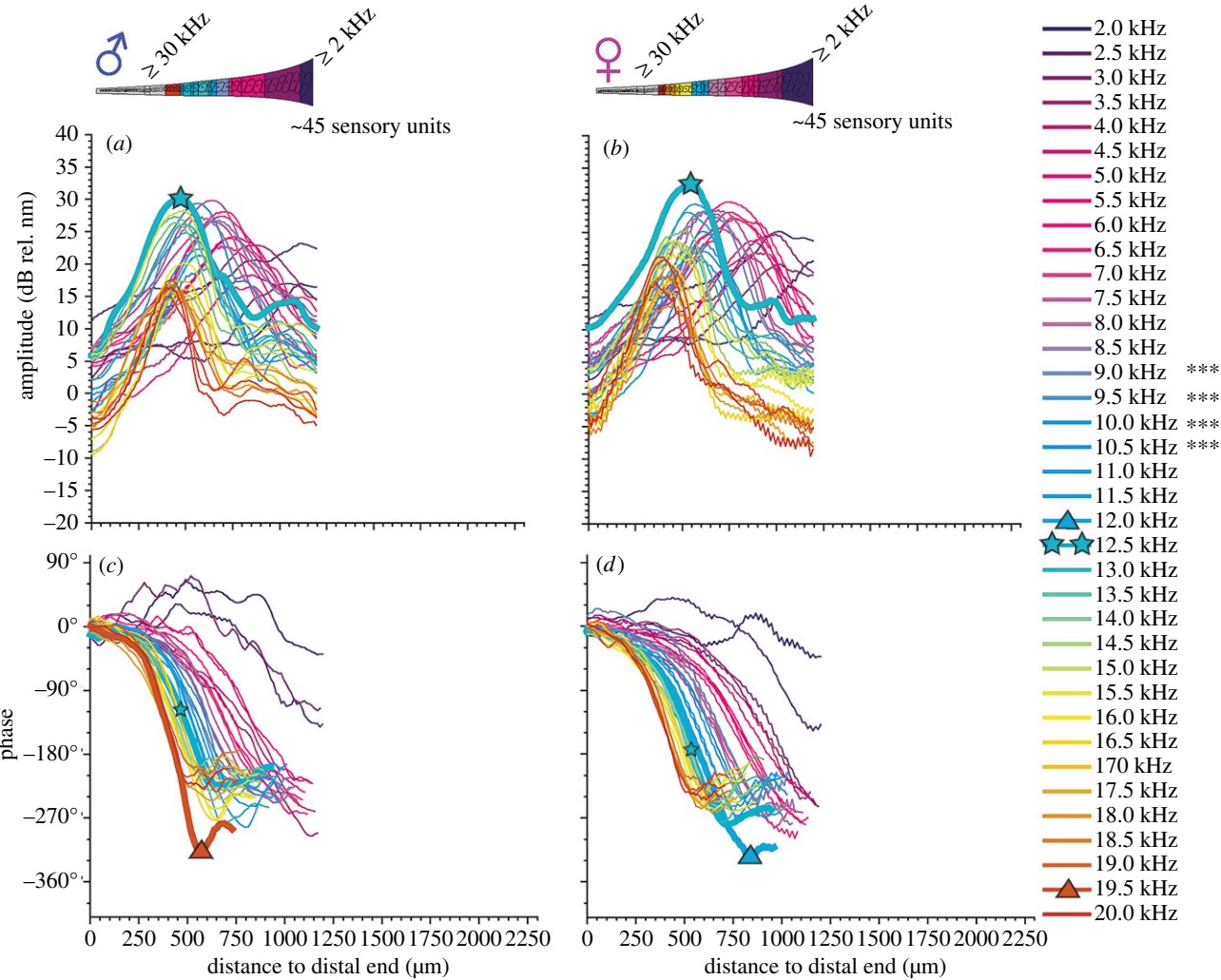

**Figure 4.** Mechanical response of the crista acustica to pure-tone stimulation in male and female *Mecopoda elongata*. Mean mechanical response to pure tones of 80 dB SPL from 2 to 20 kHz (0.5 kHz steps, see colour coded legend) was measured in male (a,c, *n* = 9) and female (b,d, *n* = 9) ears of *M. elongata*. In the medial region, at about 400–800 µm in this species, the mechanical response of the crista acustica in *M. elongata* is less sensitive to the same sound stimuli compared to *A. fenestrata* (cf. Figure 2). There is no sexual dimorphism in the crista acustica of *M. elongata*, resulting in similar mechanical properties of their respective ears. Amplitude responses are highest for 8–15 kHz stimulation in both sexes and phase delays of more than 270° are less prominent. The mechanical deflection amplitudes were significantly lower in *M. elongata* at 9–10.5 kHz in comparison to the responses in the male ears of *A. fenestrata* (*** $p < 0.001$, two-way ANOVA; same colour code and symbols for sound frequencies as in figure 3). Large stars = highest organ defection (organ deflection maximum), large triangles = maximum phase delay, small stars = phase delay at the point of maximum organ deflection. (Online version in colour.)

14 kHz (electronic supplementary material, figure S2). Sharpest responses were generally found for high frequencies in the distal crista acustica of all animals. In comparison to *M. elongata*, the ears of the bushcricket species *A. fenestrata* are obviously longer and bear more sensory cells. To answer the question of whether more sensory cells are likely to be moved by the same sound stimulation due to the higher mechanical crista acustica deflection in the fovea species *A. fenestrata*, we calculated the area under curve for each measurement and averaged these values for each frequency, species and sex (figure 5; electronic supplementary material, figure S1A). With $68.9 \, \mu m^2 \pm 30.1 \, \mu m^2$ (*n* = 9), the highest area value was determined in *A. fenestrata* males within the fovea region for 9.5 kHz sound stimulation (figure 5, left side, green line). In the frequency range of 8.0–10.5 kHz, the stimulation lead to significant higher ($p < 0.05$ to $p < 0.001$, two-way ANOVA) area motion in comparison to low-frequency (less than 6 kHz) and high-frequency (greater than 12.5 kHz) stimulation. Female *A. fenestrata* had the highest area values for 11 kHz

stimulation. This area value was only half as large as the corresponding responses in the males ($25.7 \, \mu m^2 \pm 11.2 \, \mu m^2$, *n* = 14; figure 5, left side, blue star). Comparing the data of males and females, the area under curve was significantly higher in males for sound frequencies between 9.0 kHz and 10.5 kHz ($p < 0.001$, two-way ANOVA, *n* = 9 males and 14 females). In *M. elongata*, the bushcricket without the fovea region, the calculated areas values were similar between sexes and reach values of only $14.4 \, \mu m^2 \pm 11.8 \, \mu m^2$ (8 kHz) and $14.0 \, \mu m^2 \pm 9.0 \, \mu m^2$ (12.5 kHz) in males and females, respectively (figure 5, right side). These values were significantly lower ($p < 0.003–0.001$, two-way ANOVA, *n* = 9 males and 9 females) compared to the displacement areas calculated for *A. fenestrata* males. After correcting the area under curve for the different organ lengths, the much higher values in the foveal region of male *A. fenestrata* are still obvious (figure 5b). That emphasizes the fact that not only the organ length is the major factor for increased area (sensory cells) motion, but also the fovea mechanics itself lead to higher area motion.

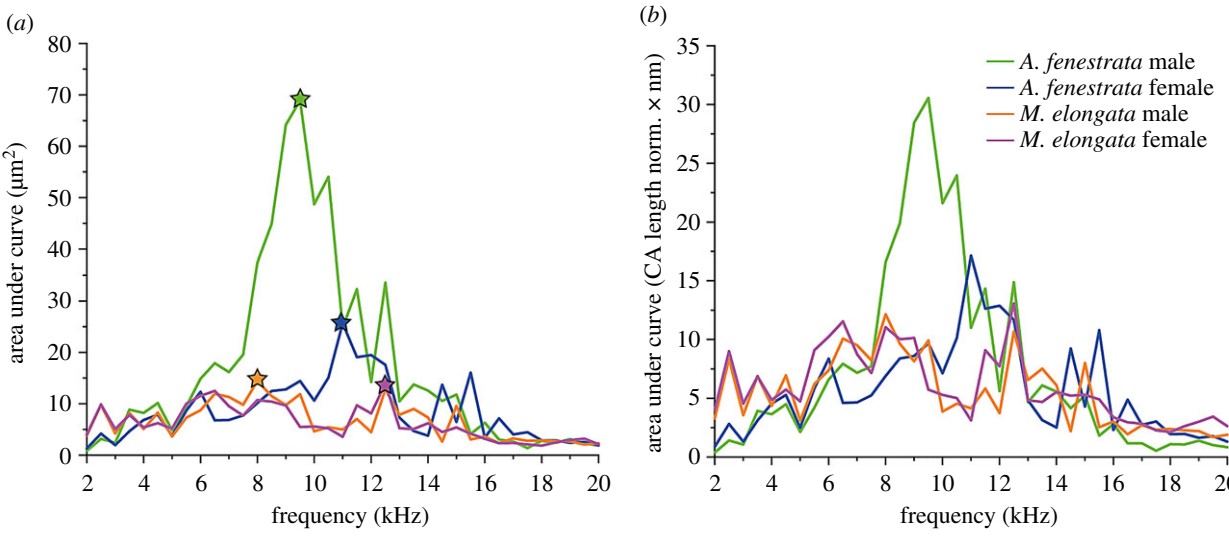

**Figure 5.** Area under curve analysis of the mechanical crista acustica response. (*a*) Area under curve calculation using organ deflection under pure tone stimulation (2–20 kHz, 80 dB SPL). Average highest area under curve value are marked by stars. (*b*) Area under curve normalized to the different crista acustica lengths. Male *A. fenestrata* still show larger area values (more sensory cells) that are moved in the foveal region. Coloured asterisks indicate the respective maxima. (Online version in colour.)

## 4. Discussion

In convergent evolution to the mammalian cochlea, bushcrickets also use sound-induced travelling waves along their hearing organs for signal transduction from sound signals to neuronal responses [32,36]. For the mechano-electrical transduction process, a pronounced phase delay that forms the travelling wave along the hearing organ is needed to open the transduction channels in the sensory cells [31]. It was unknown how an auditory fovea with suspended gradient of stiffness affects the organ mechanics and further transduction. Therefore, in this comparative study, we investigated in detail the sound-induced organ deflection in bushcricket species with (*A. fenestrata*) and without (*M. elongata*) an auditory fovea. Additionally, in *A. fenestrata*, the auditory fovea is sex-specific and most distinctly developed in male ears [19], where we found the most pronounced phase delay. This larger phase delay in the male ears was measured at the foveal region that processes the female response call frequency of about 10 kHz [16]. In the male ears, we further measured significantly higher displacement amplitudes and a larger area with sensory cells that is moved by sound stimulation (area of male-specific expanded auditory fovea). These differences become particularly obvious when comparing the sound-induced motions with *M. elongata* that does not have a fovea. We showed that the mechanical organ deflection and broadness of oscillation waves, induced by airborne sound, is remarkably higher in *A. fenestrata*. These mechanical findings in the male ear of *A. fenestrata* depend on the one hand on the elongation of the crista acustica. However, after correcting for this anatomical factor, it shows that the fovea itself enhances the area that is stimulated by sound. Therefore, we hypothesize that elongated hearing organs show more pronounced travelling waves and the presence of an auditory fovea itself enhances the organ deflection and therefore improves the mechano-electrical signal transduction within the ear. A clear plateau in the response amplitude formed from a single frequency over a longer distance of the crista acustica was not found. We attribute this finding to the small size of the total hearing organ of about 2 mm.

Sound-induced phase delays are also reported in vertebrate ears where, depending on the stimulus sound frequency, they frequently exceed one cycle [37–39]. Most work about the peripheral auditory fovea had been done in bats [40–48]. In the auditory fovea of bats, additional to the travelling wave, resonance of the tectorial membrane measured with distortion product otoacoustic emissions, is associated with an further sudden phase delay of 360° [44]. In bats, the auditory fovea is composed of a sparsely innervated region (SI) of increased basilar membrane thickness and a highly modified tectorial membrane that allows vibrations since it is loosely attached to the spiral limbus [34]. The SI region provides constant organ stiffness induced by constant height and is followed by a highly innervated straight basilar membrane region (SR). In addition to the conventional travelling wave, it was suggested that the SI region features standing waves and larger resonant tectorial membrane movements [46,49]. This standing wave resonance of the tectorial membrane drives the basilar membrane in the SI region that is enhanced, finely modulated and controlled by outer hair cells' electromotility. In bats, a disruption of the anatomical gradient is proposed to generate standing wave resonances that pre-process signals represented more apically in addition to conventional travelling waves [44]. The auditory fovea in the bushcricket *A. fenestrata* is also generated by an area of a halted anatomical gradient in the medial crista acustica [16,19]. It was shown that in bushcricket ears the mechanical response is stiffness dependent [50] and anatomical features like the dendritic height correspond to mechanical changes in frequency responses [51]. Mechanical anchors along the dendrites, called ciliary roots, contribute to the found stiffness changes. In the vertebrate cochlea, the number of spiral turns and cilia height are good indicators of the hearing range [52,53], because they determine organ stiffness. However, standing wave motion, induced by a halted anatomical gradient, in the insect auditory fovea would contradict previous findings. Hummel and colleagues showed that a pronounced phase delay by a travelling wave is needed to open transduction channels since standing wave motion without phase delay are not

suitable to open transduction channels [31]. Therefore, the higher phase delay in the fovea region of the crista acustica observed in this study supports the idea that the transduction channel near the tip of the ciliated dendrite need to be tilted to increase the open probability of the transduction channel. In mammals, different tilting points of the organ of Corti structure lead to a transformation from a longitudinal up and down motion to a shear motion that tilts V- or W-shaped hair bundles [38]. Therefore, standing wave motion in the mammalian cochlea is suitable to induce a shear force that open transduction channels.

Bushcricket ears are often specialized to match habitat conditions and specific behavioural tasks, whether it is to avoid predators by acoustic detection [54] or for intraspecific communication [55,56]. Duetting bushcrickets, like *A. fenestrata*, have more scolopidial units in their respective hearings organs in general, compared to sister species with other mate-finding strategies [57]. Males of *A. fenestrata* call for females with broadband signals, with the main frequency at about 30 kHz. There is no adaptation of the female ear to this main frequency. Therefore, the total frequency components and temporality seem to be sufficient for the females for species identification. The short and narrowband female response call in duetting bushcrickets [12] is probably causing high evolutionary pressure, resulting in adapting the amount of sensory units used to localize the sound source.

The task of mate finding is challenging in *A. fenestrata* due to short and sparse answers about 40 ms in length of the female timed about 150 ms after the male call [16]. In *A. fenestrata*, the auditory fovea seems to be useful in that the increased number of sensory units provides additional auditory input for further signal processing. By disrupting the anatomical gradient of the crista acustica, an increased phase delay improves signal transduction, which could influence the processing of interaural intensity differences in the higher processing areas [58]. More responding receptor units can facilitate spatial orientation [59] and help locating the female.

Data accessibility. The data of all shown results can be found in the electronic supplementary material, ordered by their appearance in the figures.

Authors' contributions. J.S. carried out design and evaluation of laser-Doppler vibrometry and the study, carried out statistical analyses, drafted and wrote the MS; M.N. design and wrote MS, R.T. carried out CT scans and 3D reconstruction. A.S. support micro-CT scans; All authors gave final approval for publication and agree to be held accountable for the work performed therein.

Competing interests. We declare we have no competing interests.

Funding. This work was supported by the German Research Foundation (grant no. DFG NO 841/8-1 and DFG NO 841/10-1).

Acknowledgements. We thank Adrian Richter for his support for the micro-CT leg preparations. Stefan Schöneich for helpful improvement of the manuscript.

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
