## [Reviewer comments · Proceedings of the Royal Society B: Biological Sciences]

Review History

RSPB-2020-0909.R0 (Original submission)

Review form: Reviewer 1

Recommendation

Accept with minor revision (please list in comments)

Scientific importance: Is the manuscript an original and important contribution to its field?

Excellent

General interest: Is the paper of sufficient general interest?

Good

Quality of the paper: Is the overall quality of the paper suitable?

Excellent

Is the length of the paper justified?

Yes

Should the paper be seen by a specialist statistical reviewer?

No

Do you have any concerns about statistical analyses in this paper? If so, please specify them explicitly in your report.

No

It is a condition of publication that authors make their supporting data, code and materials available - either as supplementary material or hosted in an external repository. Please rate, if applicable, the supporting data on the following criteria.

Is it accessible?

Yes

Is it clear?

Yes

Is it adequate?

Yes

Do you have any ethical concerns with this paper?

No

Comments to the Author

"Comparative micromechanics of bushcricket ears with and without a specialized auditory fovea region in the crista acustica"

This study investigates the hearing mechanisms in the ear of insects for highly complex adaptations of frequency analysis. The manuscript elaborates the novel description from this research group of an insect auditory fovea in a duetting bushcricket species, by relating the mechanical features of the fovea to a hearing organ in another species with a linear frequency representation.

Using high-resolution micromechanical techniques and microanatomy by μ CT analysis, the obtained data reported here show how the auditory fovea overcomes the constraints resulting from suspended linear tonotopic frequency representation in the hearing organ by enhanced mechanical response properties and phase delays over the auditory neurons for sensory activation.

The presentation in the manuscript is concise for this complex topic, and the figures summarise considerable amounts of data in a clear way. The work sets standards investigating in depths the diversity and adaptation of hearing organs in insects and highlights the evolutionary force on complex sensory systems by the communication systems. Some general and specific aspects should be addressed to explain the results further:

General comments:

The calls' frequency spectrum (males) and certainly also the hearing range of *A. fenestrata* extends into ultrasonic ranges. Explain why you did not use sound stimuli beyond 20 kHz here, and why the hearing of females has the maximal response amplitude so notably lowered? From the data presented with the maximal response evoked in the female hearing organ at 15.5 kHz, this indicates a mis-match to the dominant frequency of 30 kHz for male calls (l. 76). Why could such a notable difference be maintained in otherwise fine-tuned hearing organs?

Relating to this, in female *A. fenestrata*, the frequencies evoking maximal amplitudes and phase delay diverge, in contrast to the homogenous males. Are these differences related?

The introduction repeatedly highlights the similarity of mechanisms in the bushcricket ear to mammalian/vertebrate hearing. This makes for a second comparison, one between the two insect species, and one between the insects and vertebrates, and the arguments interchange at times in the introduction. This information on vertebrates is convincingly combined in one paragraph in the discussion, and maybe it can be focused in the introduction as well, e. g. in line 104, the cited

response properties in the ear of bats are stated as only discussed (not established?), and supported by an insect reference. Maybe focus here on the properties of the insect ear to work out the hypotheses of the study, and keep the larger comparison to mammals for the discussion.

The manuscript refers to a comparative study, while it rather is a comparison of mechanisms, in particular since it concerns two species which are not picked based on their phylogenetic relationships. The reader obviously catches that the data from the two bushcrickets reveal the adaptive differences in the hearing mechanisms relating to tonotopy and to the auditory fovea, and this could be referred to as “comparison of micromechanics” or similar.

You discuss for the auditory fovea the lack of a gradual tuning and a suspended stiffness gradient which allows the similar tuning in the fovea. For the male fovea, Fig. 2A shows some graded positioning of the maximal response amplitudes towards the distal area of the fovea – is there a certain extend of frequency discrimination still occurring? Or the other way around, why is there not a plateau in the response amplitude formed from a single frequency over a longer distance of the crista acustica? Could this be expected or excluded in the structure of the auditory fovea?

Specific comments:

- l. 28 What do you mean by “some species” here, bushcrickets, or insects more generally?
- l. 31 This is not an inaccuracy, but “sensory epithelium” seems not to capture the complexity of the crista acustica.
- l. 40 Higher mechanical amplitudes – in what respect?
- l. 41 oscillations of what?
- l. 62 This reference is not identifying species-specificity of acoustic signals in bushcrickets. For *M. elongata elongata*, consider also Liu et al. 2019. Three new species of genus *Mecopoda* Serville, 1831 from China (Orthoptera: Tettigoniidae: Mecopodinae). *Zootaxa* 4585(3):561 – 572
- l. 67 explain briefly the elytron, as the term is introduced here
- l.82 Maybe add some references here for the usual lack of sex-specific adaptations in insects
- l. 84 what do you mean by exceptional?
- l. 96 The tonotopic organisation could be moved up, e.g. to line 79, where the crista acustica is described in more detail, because this is a central concept that affects the concept of the auditory fovea.
- l. 96 What does the Bekesy reference address here, insects or rather mammals? If the latter, move it up front in the sentence.
- l. 102 This seems not to be a “however” argument – use “so far”?
- l. 104 introduce the phase delay here in some more detail for the broader readership who may not be familiar with it
- l. 132 The forelegs of a male from each species?
- l. 184 ...in *A. fenestrata* males...
- l. 185/186 In both species and sexes, there was a clear tonotopic distribution...
- l. 216 ...equally for 12.5 kHz...?
- l. 224 Table 1: female *M. elongata* (heading)
- l. 258 suspended gradient of stiffness?
- l. 265 significantly higher displacement
- l. 295 anchors
- l. 303 either use “these previous findings”, if reference is the one from the sentence before, or include further references relevant here
- l. 311 Is it worth including if the auditory fovea is also present in other Phaneropterinae?
- l. 348 sensory cells
- l. 349 lateral tympana - and should it be cuticula here?
- l. 355 the specimen identification is not required here - you state in the results that responses were similar for all individuals.
- l. 356 Quickly explain what you show in the schematic.
- l. 361 these examples
- l. 370 specify if the numbers give mean values etc.

1. 372 Why is there more than one thick line (indicating the maximum response amplitude) in females, which do not reach the same level?

Review form: Reviewer 2 (Kaveri Rajaraman)

Recommendation

Accept with minor revision (please list in comments)

Scientific importance: Is the manuscript an original and important contribution to its field?

Good

General interest: Is the paper of sufficient general interest?

Acceptable

Quality of the paper: Is the overall quality of the paper suitable?

Excellent

Is the length of the paper justified?

Yes

Should the paper be seen by a specialist statistical reviewer?

No

Do you have any concerns about statistical analyses in this paper? If so, please specify them explicitly in your report.

No

It is a condition of publication that authors make their supporting data, code and materials available - either as supplementary material or hosted in an external repository. Please rate, if applicable, the supporting data on the following criteria.

Is it accessible?

Yes

Is it clear?

Yes

Is it adequate?

Yes

Do you have any ethical concerns with this paper?

No

Comments to the Author

I think that the edits I have suggested are quite minor, relating to grammar and phrasing rather than anything related to the internal validity of the paper. With those edits it will be an elegant piece of work which I would recommend for publishing.

Minor edits:

line 42: correct to males of the auditory fovea possessing species A fenestrated

line 99: correct presumable to presumably

line 155: might make more sense to the reader if it is clear that the 80 dB SPL at the ipsilateral spiracle is produced by the broad band speaker; otherwise to describe the resulting SPL before introducing the broadband speaker doesn't make too much sense. Suggested language:

For acoustic stimulation, pure tones from 2 to 20 kHz (0.5 kHz steps) were produced by a function generator (NI 611x; Polytec), adjusted in SPL (audio attenuator 350D; HP and amplified (RB- Rotel) to drive a broad-band speaker (R2904/700000; ScanSpeak) that was placed about 30 cm from and perpendicular to the ipsilateral spiracle, such that the SPL at the spiracle was 80 dB SPL

line 256: correct "phase delay that form the traveling wave" to "phase delay that forms a traveling wave"

line 264: correct "of the about 10 kHz" to "of about 10 kHz"

line 272: correct to "fovea itself enhances the area"

line 273: I am not sure a comparison between *A. fenestrata* and *M. elongata* can by itself lead to a causal conclusion that elongating a hearing organ can lead to more pronounced traveling waves because many other material differences that might exist between these two ear types may not be controlled for. Might be better to state that "we hypothesize that elongated hearing organs might show more pronounced traveling waves" ..

Likewise, an auditory fovea is in some sense mechanically defined by the enhanced deflection, yes? Or just by the length of membrane upon which receptors tuned to a certain frequency sit?

line 281: correct "resonance..are associated" to "is associated"

line 371: implies intentional use: suggest changing "seems to be mainly used to increase the number of sensory units that provide.." to "seems to be useful in that the increased number of sensory units provide..."

-

NOTE: the figure legends in the manuscript don't match the legends that come with the figures themselves. There are many mistakes in the latter but not the former.

Figure 2 legend: correct spelling of vibrometry from vibromerty.

In both the legend and figure sections, in the last sentence, In the last sentence, replace in "this examples" with "these examples" or "this example"

+ Also in that last line, where it says medial part ... for *A. fenestrata*, it doesn't list which part for *M. elongata*? "and (A) and in the part with 7 kHz stimulation should change -> in the medial? part (of the ear?)" (It looks medial, but more distal than other higher frequency responses?)

Figure 3 legend: legend in text is missing the relative length of male and female ears of *A. fenestrata* (whereas this is in the legend below the figure).

Also at (up to 2.25 mm and 1.8 mm, respectively - also this needs an end bracket): need to add ")" + (red shaded area of higher phase delays compared to females or *M. elongata*) should change to (...females of? *M. elongata*)

+ In legend under figure, a large triangles -> a large triangle

In the last line, correct "significant higher" to "significantly higher"

Figure 4:

In legend under figure (but not in text), correct "the mechanical amplitudes were significant lower" to "significantly lower"

+ In the last line under the figure, correct large stars = heights organ defection -> organ defection maximum, and small stars = phase delay at the point of maximum organ deflection?

Figure 5 legend: more sensory cells is extrapolated from higher areas of the crista acustica? Why not look above 30 kHz for the possible presence of an auditory fovea in that range for females?

Decision letter (RSPB-2020-0909.R0)

24-May-2020

Dear Professor Nowotny

I am pleased to inform you that your manuscript RSPB-2020-0909 entitled "Comparative micromechanics of bushcricket ears with and without a specialized auditory fovea region in the crista acustica" has been accepted for publication in Proceedings B. Congratulations!!

The referee(s) have recommended publication, but also suggest some minor revisions to your manuscript. Therefore, I invite you to respond to the referee(s)' comments and revise your manuscript. Because the schedule for publication is very tight, it is a condition of publication that you submit the revised version of your manuscript within 7 days. If you do not think you will be able to meet this date please let us know.

- DNA sequences: Genbank accessions F234391-F234402

- Phylogenetic data: TreeBASE accession number S9123
- Final DNA sequence assembly uploaded as online supplemental material
- Climate data and MaxEnt input files: Dryad doi:10.5521/dryad.12311

[http://datadryad.org/submit?journalID=RSPB&manu=\(Document not available\)](http://datadryad.org/submit?journalID=RSPB&manu=(Document+not+available)) which will take you to your unique entry in the Dryad repository. If you have already submitted your data to dryad you can make any necessary revisions to your dataset by following the above link. Please see <https://royalsociety.org/journals/ethics-policies/data-sharing-mining/> for more details.

Sincerely,

Dr John Hutchinson, Editor
mailto: proceedingsb@royalsociety.org

Associate Editor

Board Member: 1

Comments to Author:

Thank you for the opportunity to read this paper. The authors compare the micromechanics of auditory fovea in bushcrickets with and without enlargement of this structure. The differences they find in these species leads them to speculate on the adaptive advantages of enlargement of the auditory fovea. This work is outside my immediate area of expertise, but nonetheless I found it interesting (though difficult to digest in places as a non-expert) and to my relatively naïve eyes the study seemed highly novel and to deliver interesting and at least moderately far reaching conclusions. These positive views are shared by the expert reviewers. Nevertheless, both reviewers do make a number of criticisms and recommendations. It is my view that the authors should be invited to respond to reviews and submit a revised version to Proceedings B that addresses them.

Reviewer(s)' Comments to Author:

Referee: 1

Comments to the Author(s)

"Comparative micromechanics of bushcricket ears with and without a specialized auditory fovea region in the crista acustica"

This study investigates the hearing mechanisms in the ear of insects for highly complex adaptations of frequency analysis. The manuscript elaborates the novel description from this research group of an insect auditory fovea in a duetting bushcricket species, by relating the mechanical features of the fovea to a hearing organ in another species with a linear frequency representation.

Using high-resolution micromechanical techniques and microanatomy by μ CT analysis, the obtained data reported here show how the auditory fovea overcomes the constraints resulting from suspended linear tonotopic frequency representation in the hearing organ by enhanced mechanical response properties and phase delays over the auditory neurons for sensory activation.

The presentation in the manuscript is concise for this complex topic, and the figures summarise considerable amounts of data in a clear way. The work sets standards investigating in depths the diversity and adaptation of hearing organs in insects and highlights the evolutionary force on complex sensory systems by the communication systems. Some general and specific aspects should be addressed to explain the results further:

General comments:

The calls' frequency spectrum (males) and certainly also the hearing range of *A. fenestrata* extends into ultrasonic ranges. Explain why you did not use sound stimuli beyond 20 kHz here, and why the hearing of females has the maximal response amplitude so notably lowered? From the data presented with the maximal response evoked in the female hearing organ at 15.5 kHz, this indicates a mis-match to the dominant frequency of 30 kHz for male calls (l. 76). Why could such a notable difference be maintained in otherwise fine-tuned hearing organs?

Relating to this, in female *A. fenestrata*, the frequencies evoking maximal amplitudes and phase delay diverge, in contrast to the homogenous males. Are these differences related?

The introduction repeatedly highlights the similarity of mechanisms in the bushcricket ear to mammalian/vertebrate hearing. This makes for a second comparison, one between the two insect species, and one between the insects and vertebrates, and the arguments interchange at times in the introduction. This information on vertebrates is convincingly combined in one paragraph in the discussion, and maybe it can be focused in the introduction as well, e. g. in line 104, the cited response properties in the ear of bats are stated as only discussed (not established?), and supported by an insect reference. Maybe focus here on the properties of the insect ear to work out the hypotheses of the study, and keep the larger comparison to mammals for the discussion.

The manuscript refers to a comparative study, while it rather is a comparison of mechanisms, in particular since it concerns two species which are not picked based on their phylogenetic relationships. The reader obviously catches that the data from the two bushcrickets reveal the adaptive differences in the hearing mechanisms relating to tonotopy and to the auditory fovea, and this could be referred to as "comparison of micromechanics" or similar.

You discuss for the auditory fovea the lack of a gradual tuning and a suspended stiffness gradient which allows the similar tuning in the fovea. For the male fovea, Fig. 2A shows some graded positioning of the maximal response amplitudes towards the distal area of the fovea – is there a certain extend of frequency discrimination still occurring? Or the other way around, why is there not a plateau in the response amplitude formed from a single frequency over a longer distance of the crista acustica? Could this be expected or excluded in the structure of the auditory fovea?

Specific comments:

- l. 28 What do you mean by "some species" here, bushcrickets, or insects more generally?
- l. 31 This is not an inaccuracy, but "sensory epithelium" seems not to capture the complexity of the crista acustica.
- l. 40 Higher mechanical amplitudes – in what respect?
- l. 41 oscillations of what?
- l. 62 This reference is not identifying species-specificity of acoustic signals in bushcrickets. For *M. elongata elongata*, consider also Liu et al. 2019. Three new species of genus *Mecopoda* Serville, 1831 from China (Orthoptera: Tettigoniidae: Mecopodinae). *Zootaxa* 4585(3):561 – 572
- l. 67 explain briefly the elytron, as the term is introduced here
- l.82 Maybe add some references here for the usual lack of sex-specific adaptations in insects
- l. 84 what do you mean by exceptional?

- l. 96 The tonotopic organisation could be moved up, e.g. to line 79, where the crista acustica is described in more detail, because this is a central concept that affects the concept of the auditory fovea.
- l. 96 What does the Bekesy reference address here, insects or rather mammals? If the latter, move it up front in the sentence.
- l. 102 This seems not to be a “however” argument – use “so far”?
- l. 104 introduce the phase delay here in some more detail for the broader readership who may not be familiar with it
- l. 132 The forelegs of a male from each species?
- l. 184 ...in *A. fenestrata* males was...
- l. 185/186 In both species and sexes, there was a clear tonotopic distribution...
- l. 216 ...equally for 12.5 kHz...?
- l. 224 Table 1: female *M. elongata* (heading)
- l. 258 suspended gradient of stiffness?
- l. 265 significantly higher displacement
- l. 295 anchors
- l. 303 either use “these previous findings”, if reference is the one from the sentence before, or include further references relevant here
- l. 311 Is it worth including if the auditory fovea is also present in other *Phaneropterinae*?
- l. 348 sensory cells
- l. 349 lateral tympana - and should it be cuticula here?
- l. 355 the specimen identification is not required here - you state in the results that responses were similar for all individuals.
- l. 356 Quickly explain what you show in the schematic.
- l. 361 these examples
- l. 370 specify if the numbers give mean values etc.
- l. 372 Why is there more than one thick line (indicating the maximum response amplitude) in females, which do not reach the same level?

Referee: 2

Comments to the Author(s)

I think that the edits I have suggested are quite minor, relating to grammar and phrasing rather than anything related to the internal validity of the paper. With those edits it will be an elegant piece of work which I would recommend for publishing.

Minor edits:

line 42: correct to males of the auditory fovea possessing species *A. fenestrata*

line 99: correct presumable to presumably

line 155: might make more sense to the reader if it is clear that the 80 dB SPL at the ipsilateral spiracle is produced by the broad band speaker; otherwise to describe the resulting SPL before introducing the broadband speaker doesn't make too much sense. Suggested language: . For acoustic stimulation, pure tones from 2 to 20 kHz (0.5 kHz steps) were produced by a function generator (NI 611x; Polytec), adjusted in SPL (audio attenuator 350D; HP and amplified (RB-Rotel) to drive a broad-band speaker (R2904/700000; ScanSpeak) that was placed about 30 cm from and perpendicular to the ipsilateral spiracle, such that the SPL at the spiracle was 80 dB SPL

line 256: correct “phase delay that form the traveling wave” to “phase delay that forms a traveling wave”

line 264: correct “of the about 10 kHz” to “of about 10 kHz”

line 272: correct to “fovea itself enhances the area”

line 273: I am not sure a comparison between *A. fenestrata* and *M. elongata* can by itself lead to a causal conclusion that elongating a hearing organ can lead to more pronounced traveling waves because many other material differences that might exist between these two ear types may not be controlled for. Might be better to state that “we hypothesize that elongated hearing organs might show more pronounced traveling waves”..

Likewise, an auditory fovea is in some sense mechanically defined by the enhanced deflection, yes? Or just by the length of membrane upon which receptors tuned to a certain frequency sit?
 line 281: correct "resonance..are associated" to "is associated"
 line 371: implies intentional use: suggest changing "seems to be mainly used to increase the number of sensory units that provide.." to "seems to be useful in that the increased number of sensory units provide.."

-
 NOTE: the figure legends in the manuscript don't match the legends that come with the figures themselves. There are many mistakes in the latter but not the former.

Figure 2 legend: correct spelling of vibrometry from vibromerty.

In both the legend and figure sections, in the last sentence, In the last sentence, replace in "this examples" with "these examples" or "this example"

+ Also in that last line, where it says medial part ... for A fenestrata, it doesn't list which part for M elongata? "and (A) and in the part with 7 kHz stimulation should change -> in the medial? part (of the ear?)" (It looks medial, but more distal than other higher frequency responses?)

Figure 3 legend: legend in text is missing the relative length of male and female ears of A. fenestrata (whereas this is in the legend below the figure).

Also at (up to 2.25 mm and 1.8 mm, respectively - also this needs an end bracket): need to add ")"

+ (red shaded area of higher phase delays compared to females or M. elongata) should change to (..females of? M. elongata)

+ In legend under figure, a large triangles -> a large triangle

In the last line, correct "significant higher" to "significantly higher"

Figure 4:

In legend under figure (but not in text), correct "the mechanical amplitudes were significant lower" to "significantly lower"

+ In the last line under the figure, correct large stars = heights organ deflection -> organ deflection maximum, and small stars = phase delay at the point of maximum organ deflection?

Figure 5 legend: more sensory cells is extrapolated from higher areas of the crista acustica? Why not look above 30 kHz for the possible presence of an auditory fovea in that range for females?

Author's Response to Decision Letter for (RSPB-2020-0909.R0)

See Appendix A.

Decision letter (RSPB-2020-0909.R1)

03-Jun-2020

Dear Professor Nowotny

I am pleased to inform you that your manuscript entitled "Comparative micromechanics of bushcricket ears with and without a specialized auditory fovea region in the crista acustica" has been accepted for publication in Proceedings B.

Open Access

Paper charges

Sincerely,

Proceedings B

Appendix A

Response to Referee: 1

General comments:

The calls' frequency spectrum (males) and certainly also the hearing range of *A. fenestrata* extends into ultrasonic ranges. Explain why you did not use sound stimuli beyond 20 kHz here, and why the hearing of females has the maximal response amplitude so notably lowered? From the data presented with the maximal response evoked in the female hearing organ at 15.5 kHz, this indicates a mis-match to the dominant frequency of 30 kHz for male calls (l. 76). Why could such a notable difference be maintained in otherwise fine-tuned hearing organs?

Response: The male call and female response is different in frequency but also in broadness. Males call with a more broadband signal and females respond with a short pulse and precise time window and narrowband signal. Therefore, we do not think that this is a mismatch. We wrote in the discussion (line 316): *Males of A. fenestrata, call for females with broadband signals, with the main frequency at about 30 kHz. There is no adaptation of the female ear to this main frequency. Therefore, the total frequency components and temporality seem to be sufficient for the females for species identification. The short and narrowband female response call...*

Relating to this, in female *A. fenestrata*, the frequencies evoking maximal amplitudes and phase delay diverge, in contrast to the homogenous males. Are these differences related?

Response: We interpret this difference in the context that females also form a small fovea at 10 kHz. We can show this very clearly in anatomical studies of the organs (Scherberich et al 2017, Fig. 3). How these processes are to be evaluated in evolutionary terms, is still too speculative for us. Therefore, we have not investigated this difference further in this study but this is something which is planned for the future.

The introduction repeatedly highlights the similarity of mechanisms in the bushcricket ear to mammalian/vertebrate hearing. This makes for a second comparison, one between the two insect species, and one between the insects and vertebrates, and the arguments interchange at times in the introduction. This information on vertebrates is convincingly combined in one paragraph in the discussion, and maybe it can be focused in the introduction as well, e. g. in line 104, the cited response properties in the ear of bats are stated as only discussed (not established?), and supported by an insect reference. Maybe focus here on the properties of the insect ear to work out the hypotheses of the study, and keep the larger comparison to mammals for the discussion.

Response: So far, only results from the cochlea of mammals are available that explain the mechanics behind an auditory fovea. But we have reduced the comparison to the bats to keep the reading better. We now write in the introduction (line 104): *A foveal standing wave in the bushcricket ear, as found in mammals [27], would contradict our previous findings in *M. elongata*, which indicate the need of a phase change as mechanical basis of signal transduction.*

The manuscript refers to a comparative study, while it rather is a comparison of mechanisms, in particular since it concerns two species which are not picked based on their phylogenetic relationships. The reader obviously catches that the data from the two bushcrickets reveal the adaptive differences in the hearing mechanisms relating to tonotopy and to the auditory fovea, and this could be referred to as "comparison of micromechanics" or similar.

Response: At the beginning of the results part we have clarified the different group classifications and write in the text (line 181): *...and measured the displacement amplitude and phase responses of the crista acustica in bushcrickets with (*Ancylecha fenestrata*) and without (*Mecopoda elongata*)*

an auditory fovea that belong the different subfamilies (Phaneropterinae and Mecopodinae, respectively).

You discuss for the auditory fovea the lack of a gradual tuning and a suspended stiffness gradient which allows the similar tuning in the fovea. For the male fovea, Fig. 2A shows some graded positioning of the maximal response amplitudes towards the distal area of the fovea – is there a certain extend of frequency discrimination still occurring? Or the other way around, why is there not a plateau in the response amplitude formed from a single frequency over a longer distance of the crista acustica? Could this be expected or excluded in the structure of the auditory fovea?

Response: Bushcrickets have very short hearing organs in comparison to mammals. The ears of *A. fenestra* males are with 2 mm unusually long. That could be the reason why it is not possible to create a plateau but slowdown the frequency tuning. However, we found an increase of the amplitudes due to the auditory fovea. This can be seen in the significant differences to the females of *A. fenestrata* and the males and females of *M. elongata*. Which is further supported by the increased values in the AUC measurements. To make this point clear in the text, we now write in the discussion (line 276): *A clear plateau in the response amplitude formed from a single frequency over a longer distance of the crista acustica was not found. We attribute this finding to the small size of the total hearing organ of about 2 mm.*

Specific comments:

I. 28 What do you mean by “some species” here, bushcrickets, or insects more generally?

Response: We mean more generally. We changed the wording to: *In some insects and vertebrate species,...*

I. 31 This is not an inaccuracy, but “sensory epithelium” seems not to capture the complexity of the crista acustica.

Response: We thank the reviewer for pointing to this, but at this point in the paper we only intend to introduce into the topics. We address and explain the complexity of the *crista acustica* later in the discussion part.

I. 40 Higher mechanical amplitudes – in what respect?

Response: We write now: *...leads to higher mechanical amplitudes and longer phase delays in A. fenestrata male ears.*

I. 41 oscillations of what?

Response: We write now: *... analyses of the organ oscillations reveal.*

I. 62 This reference is not identifying species-specificity of acoustic signals in bushcrickets.

For *M. elongata elongata*, consider also Liu et al. 2019. Three new species of genus *Mecopoda* Serville, 1831 from China (Orthoptera: Tettigoniidae: Mecopodinae). *Zootaxa* 4585(3):561 – 572

Response: We included the mentioned publication by Liu et al. 2019

I. 67 explain briefly the elytron, as the term is introduced here

Response: We write now: *The female stridulatory organs always consist of small teeth on the dorsal surface of the right elytron (front wing protecting the hind wings) but show distinct variation among different species [15].*

I.82 Maybe add some references here for the usual lack of sex-specific adaptations in insects

Response: There is not much literature about the lack of sex-specific adaptations in ears, but we added examples of some cases where it is known (line 82).

I. 84 what do you mean by exceptional?

Response: We removed the word exceptional.

I. 96 The tonotopic organisation could be moved up, e.g. to line 79, where the crista acustica is described in more detail, because this is a central concept that affects the concept of the auditory fovea.

Response: We describe in line 77 that the ear is tonotopically organized. It is therefore set in relation to the concept of an auditory fovea. We have kept this sentence in this position for a smoother transition to a new section.

I. 96 What does the Bekesy reference address here, insects or rather mammals? If the latter, move it up front in the sentence.

Response: We moved the Bekesy references to the up front.

I. 102 This seems not to be a “however” argument – use “so far”?

Response: We agree and changed this.

I. 104 introduce the phase delay here in some more detail for the broader readership who may not be familiar with it

Response: We write now (line 105): *In general, oscillatory organ motion in bushcrickets, can be described by an amplitude and phase component. When in the direction of the mechanical wave propagation, the phase response lag along the hearing organ, it is called a phase delay.*

I. 132 The forelegs of a male from each species?

Response: We write now: *The forelegs of two male individuals from each species were dehydrated...*

I. 184 ...in *A. fenestrata* males was...

Response: We agree and changed this.

I. 185/186 In both species and sexes, there was a clear tonotopic distribution...

Response: We agree and changed this.

I. 216 ...equally for 12.5 kHz...?

Response: We agree and changed this.

I. 224 Table 1: female *M. elongata* (heading)

Response: We agree and changed this.

I. 258 suspended gradient of stiffness?

Response: We agree and changed this.

I. 265 significantly higher displacement

Response: We agree and changed this.

I. 295 anchors

Response: We agree and changed this.

I. 303 either use “these previous findings”, if reference is the one from the sentence before, or include further references relevant here

Response: That is right. We removed the phrase “these previous findings”.

I. 311 Is it worth including if the auditory fovea is also present in other Phaneropterinae?

Response: Yes, since we do not know of any other hearing organ in bushcrickets that has more sensory cells.

I. 348 sensory cells

Response: We agree and changed this.

I. 349 lateral tympana - and should it be cuticula here?

Response: We agree and changed this.

I. 355 the specimen identification is not required here - you state in the results that responses were similar for all individuals.

Response: We agree and changed this.

I. 356 Quickly explain what you show in the schematic.

Response: We agree and explain this.

I. 361 these examples

Response: We agree and changed this.

I. 370 specify if the numbers give mean values etc.

Response: We agree and changed this.

I. 372 Why is there more than one thick line (indicating the maximum response amplitude) in females, which do not reach the same level?

Response: We agree and changed this. There is only one thick line now.

Referee: 2

Comments to the Author(s)

I think that the edits I have suggested are quite minor, relating to grammar and phrasing rather than anything related to the internal validity of the paper. With those edits it will be an elegant piece of work which I would recommend for publishing.

Minor edits:

line 42: correct to males of the auditory fovea possessing species A fenestrate

Response: We agree and changed this.

line 99: correct presumable to presumably

Response: We agree and changed this.

line 155: might make more sense to the reader if it is clear that the 80 dB SPL at the ipsilateral spiracle is produced by the broad band speaker; otherwise to describe the resulting SPL before introducing the broadband speaker doesn't make too much sense. Suggested language: . For acoustic stimulation, pure tones from 2 to 20 kHz (0.5 kHz steps) were produced by a function generator (NI 611x; Polytec), adjusted in SPL (audio attenuator 350D; HP and amplified (RB- Rotel) to drive a broad-band speaker (R2904/700000; ScanSpeak) that was placed about 30 cm from and perpendicular to the ipsilateral spiracle, such that the SPL at the spiracle was 80 dB SPL

Response: We agree and changed this.

line 256: correct "phase delay that form the traveling wave" to "phase delay that forms a traveling wave"

Response: We agree and changed this.

line 264: correct "of the about 10 kHz" to "of about 10 kHz"

Response: We agree and changed this.

line 272: correct to "fovea itself enhances the area"

Response: We agree and changed this.

line 273: I am not sure a comparison between *A. fenestrata* and *M. elongata* can by itself lead to a causal conclusion that elongating a hearing organ can lead to more pronounced traveling waves because many other material differences that might exist between these two ear types may not be controlled for. Might be better to state that "we hypothesize that elongated hearing organs might show more pronounced traveling waves"..

Response: We agree and changed this.

Likewise, an auditory fovea is in some sense mechanically defined by the enhanced deflection, yes? Or just by the length of membrane upon which receptors tuned to a certain frequency sit?

Response: The term fovea describes the representation of a physiologically important stimulus (like the frequency of a tone) on the usual half of the total organ length. It is not describing the amplitude of the deflection.

line 281: correct "resonance..are associated" to "is associated"

Response: We agree and changed this.

line 371: implies intentional use: suggest changing "seems to be mainly used to increase the number of sensory units that provide.." to "seems to be useful in that the increased number of sensory units provide..."

Response: We agree and changed this.

NOTE: the figure legends in the manuscript don't match the legends that come with the figures themselves. There are many mistakes in the latter but not the former.

Figure 2 legend: correct spelling of vibrometry from vibromerty.

Response: We agree and changed this.

In both the legend and figure sections, in the last sentence, In the last sentence, replace in “this examples” with “these examples” or “this example”

Response: We agree and changed this.

+ Also in that last line, where it says medial part ... for A fenestrata, it doesn't list which part for M elongata? “and (A) and in the part with 7 kHz stimulation should change -> in the medial? part (of the ear?)” (It looks medial, but more distal than other higher frequency responses?)

Response: We write now: *In these examples, the highest response amplitudes were recorded in the medial part of both ears with 10 kHz stimulation for A. fenestrata (A) and 7 kHz stimulation for M. elongata (B).*

Figure 3 legend: legend in text is missing the relative length of male and female ears of A. fenestrata (whereas this is in the legend below the figure).

Response: We agree and included this information.

Also at (up to 2.25 mm and 1.8 mm, respectively - also this needs an end bracket): need to add “)”

Response: We agree and changed this.

+ (red shaded area of higher phase delays compared to females or M. elongata) should change to (..females of? M. elongata)

Response: We agree and changed this.

+ In legend under figure, a large triangles -> a large triangle

Response: We agree and changed this.

In the last line, correct “significant higher” to “significantly higher”

Response: We agree and changed this.

Figure 4:

In legend under figure (but not in text), correct “the mechanical amplitudes were significant lower” to “significantly lower”

Response: We agree and changed this.

+ In the last line under the figure, correct large stars = heights organ defection -> organ defection maximum, and small stars = phase delay at the point of maximum organ defection?

Response: We agree and changed this.

Figure 5 legend: more sensory cells is extrapolated from higher areas of the crista acustica? ^{is it} Why not look above 30 kHz for the possible presence of an auditory fovea in that range for females?

Response: We know from previous studies that there is no female auditory fovea at frequencies above 20 kHz (Scherberich et al. 2015 and 2016).